# Numerical Study on the Hydrodynamic Performance of a Flexible Caudal Fin with Different Trailing-Edge Shapes

**DOI:** 10.3390/biomimetics9070445

**Published:** 2024-07-21

**Authors:** May Hlaing Win Khin, Shinnosuke Obi

**Affiliations:** 1Department of Mechanical Engineering, West Yangon Technological University, Yangon 11401, Myanmar; 2Department of Mechanical Engineering, Keio University, Yokohama 223-8522, Japan

**Keywords:** flexible caudal fin, trailing-edge shape, hydrodynamic performances

## Abstract

This paper presents a three-dimensional fluid-structure-coupled simulation of a flexible caudal fin with different trailing-edge shapes. The influences of caudal-fin shape on hydrodynamic performance are investigated by comparing the results of a simplified model of a square caudal fin with forked and deeply forked caudal fins under a wider range of non-dimensional flapping frequency, 0.6 < *f** < 1.5, where *f** is the ratio of flapping frequency to the natural frequency of each caudal fin, i.e., *f** = *f*/*f*_n_. The leading edge of each caudal fin is forced to oscillate vertically in a water tank with zero free-stream conditions. The numerical results show that the amount of forking in the geometry of the caudal fin has significant effects on its hydrodynamic performance. A comparison of thrust coefficients shows that the square caudal fin has a greater thrust coefficient in the non-dimensional frequency range of 0.6 < *f** < 1.2, while the deeply forked caudal fin generates higher thrust when 1.2 < *f** < 1.5. In terms of propulsive efficiency, the square caudal fin is more efficient when 0.6 < *f** < 0.9, while the propulsive efficiency of a deeply forked caudal fin is significantly enhanced when 0.9 < *f** < 1.5. Based on our results, the deeply forked caudal fin has greater thrust coefficients and a higher propulsive efficiency in a higher frequency range than the natural frequency of each caudal fin. The thrust characteristics and flow fields around each caudal fin are investigated in detail.

## 1. Introduction

The propulsion mechanisms of fish and aquatic animals have been attractive to researchers for many years. A large number of investigations concerning models of fish and caudal fins have been performed to examine the hydrodynamics of the flapping propulsors [1,2,3,4,5,6,7]. In addition, the unsteady flow fields induced by the oscillation of the caudal fin have been examined from various perspectives, such as shape, kinematics, and material flexibility [8,9,10,11,12,13,14]. Among these aspects, caudal-fin shape was recognized as one of the most important aspects for thrust improvement in biological locomotion. In order to clarify the effect of caudal-fin shape on thrust generation and swimming performance, a large number of investigations have been presented in the literature for various geometries of caudal fin models [15,16,17,18]. Caudal fins have a wide variety of shapes, and their shapes have been characterized by basic parameters such as aspect ratio and area moments.

Many previous studies have examined the influence of the aspect ratio of caudal fins on hydrodynamic and swimming performance [19,20,21]. Lauder et al. [22] investigated the effect of different tail shapes on swimming performance using five different flexible foils with various aspect ratios. They concluded that a caudal fin with a higher aspect ratio foil and an angled trailing edge allows for significantly faster swimming than a rectangular foil. Liu et al. [23] investigated the hydrodynamic performance and wake patterns of different caudal-fin shapes with various aspect ratios in flapping propulsion. They indicated that large-aspect-ratio caudal fins have a higher efficiency during cruising, while low-aspect-ratio caudal fins are more efficient during accelerating and maneuvering.

Matta et al. [24] examined the impact of caudal-fin shape on the thrust production of a biomimetic robotic tuna. They used three different kinds of caudal fins—rectangular, elliptical, and swept—which had the same area and aspect ratio. They showed that the swept fin had the greatest thrust potential at a high Strouhal number, followed by the elliptical fin. In the work of Li et al. [25], the propulsion performances of typical fishlike tail shapes were compared with those of a rectangular plate by setting the surface areas of the plates to be the same in all cases. Their results illustrated that the fishlike, forked plates have better locomotion performance.

Van Buren et al. [26] performed an experimental work on the rigid pitching panel to study the influence of trailing-edge shape on propulsive performance. In that work, the trailing edges of a rigid panel were shaped in concave, square, and convex forms, while the aspect ratio of the panel was kept constant. Their results indicated that changing the trailing-edge shape from concave to convex improved thrust and efficiency. Baba et al. [27] investigated the flow field around the caudal fin using different trailing-edge shapes of flexible caudal fins in fixed propulsive conditions. In that work, the shapes of the trailing edges were characterized by chord length *Lc* at the center of the caudal fin to obtain three types of caudal fin—square, forked, and deeply forked. Velocity measurements were performed in the center plane of these caudal fins, and the magnitude of the jet velocity was observed to be dependent on fin shape, where the deeply forked shape provided the strongest jet-like flow.

Rosic et al. [28] investigated the effect of kinematics and stiffness on the propulsive performance of a single foil shape and reported that these two parameters exhibited a subtle interacting effect on hydrodynamic efficiency. Zhang et al. [29] also investigated the influence of trailing-edge shape on self-propulsive performance under different bending stiffnesses. In their study, the trailing edge angles of three flexible plates were varied from 30 to 150 degrees to create square, convex and concave plates. They observed that the square and concave plates achieved the best performance under small and large bending stiffness, respectively. Their results suggest that the best performance achieved by each trailing-edge shape is dependent on the material’s flexibility. Feilich et al. [30] conducted experiments to study how shape interacts with stiffness to produce swimming performance. They used four different foil shapes made of three types of plastic that have different flexural stiffnesses. They suggested that both shape and stiffness are important in determining the propulsive performance of undulating foils and that these two parameters interact in complicated ways.

We extend the experimental work of Baba et al. [27] to three types of caudal fin with different trailing-edge shapes using numerical simulation for the fluid–structure interaction of flexible caudal fins. Since hydrodynamic performance results from the interaction of the forces induced by the deformable body and those induced by the surrounding fluid, the interaction between the fluid’s motion and the flexible caudal fin is key to understanding the mechanisms of hydrodynamic performance. The objective of this study is to achieve further understanding of the relation between caudal-fin shape and the frequency characteristics of hydrodynamic performance. All caudal fins maintain a constant length of leading edge and side edges, with the varying chord length at the center of the caudal fin. This allows the trailing edge to be different shapes; square, forked, and deeply forked caudal fins correspond to the cases 1, 2, and 3, respectively. The leading edge of a caudal fin is forced to oscillate with sinusoidal motion in a stationary fluid. The difference in hydrodynamic performance due to the change in trailing-edge shape, the vortex structures, and the corresponding pressure distributions are investigated comparatively.

## 2. Materials and Methods

The flow fields around a flapping caudal fin are numerically simulated by open-source CFD software OpenFOAM that provides a partitioned fluid–structure interaction solver *fsiFoam* in the version of foam-extend-3.2. 

### 2.1. Geometric Model

Figure 1 shows the schematic geometry of the three caudal fins with flat and forked trailing-edge shapes. These three types of trailing edges shape are selected based on the experimental results of Baba et al. [27], which provided the maximum performance in each group of caudal-fin shapes. The trailing-edge shape along the span is defined by the chord length *Lc* at the mid-span of caudal fin, as defined in the experiments. A square caudal fin with a side length of *Lx* = 40 mm and a uniform thickness of 1 mm, shown in Figure 1a, is the basic model of caudal fin and is termed “case 1”. The trailing-edge shapes are varied by decreasing the chord length *Lc* from 40 to 8 mm in intervals of 16 mm, where *Lc* = 24 mm and *Lc* = 8 mm correspond to the forked and deeply forked caudal fins shown in Figure 1b,c, respectively. All these caudal fins are of the same side length (*Lx* = 40 mm) but of different chord length *Lc* and different surface areas *S*. The forked caudal fin shown in Figure 1b is termed “case 2”, and the deeply forked caudal fin shown in Figure 1c is termed “case 3”. The shape of the caudal fin is characterized by the aspect ratio *AR* = *Lx*^2^/*S*. The geometric parameters used in this paper are summarized in Table 1. The leading edge of each caudal fin is forced to oscillate in the *z* direction as follows:*z_r_* = *A_r_* sin (2π*ϕ*), *ϕ* = *ft* (0 < *ϕ* < 1),(1)
where *z_r_* is the vertical displacement of the leading edge, and *A_r_* is the flapping amplitude of the leading edge. The term *ϕ* denotes the non-dimensional flapping phase, which is a product of flapping frequency *f* and time *t*.

### 2.2. Governing Equations

The governing equations for both fluid and solid fields can be found in our previous work [31]. The fluid flow field is governed by the incompressible Navier–Stokes equations for viscous flow. To solve the fluid flow equations on a deforming mesh, the arbitrary Lagrangian–Eulerian method is employed [32]:∇. ***u_f_*** = 0,(2)
∂***u_f_***/∂*t* + (***u_f_*** − ***u_m_,_f_***).∇ ***u_f_*** = −∇*p*/*ρ* + *υ_f_* ∇^2^ ***u_f_***(3)
where ***u_f_*** and *p* are the fluid velocity and pressure, respectively, ***u_m,f_*** is the mesh velocity, *ρ* is the fluid density, and *υ_f_* is the kinematic viscosity of the fluid. The governing equations of structural field can be described as follows [33].
*ρ*_s_ ∂^2^***u***/∂*t*^2^ = ∇.(**∑**.***F*^T^**) + *ρ*_s_ ***f*_b_**(4)
where ***u*** is the displacement of the solid, *ρ*_s_ is the density of the solid, **∑** is the second Piola–Kirchhoff stress tensor, ***F*** is the deformation gradient tensor, and ***f***_b_ is the resulting body force. The relation for the constitutive equation of a Saint Venant–Kirchhoff material and the deformation gradient tensor ***F*** is as follows:**∑** = 2*μ_s_* ***G***+ *λ_s_* tr(***G***)***I***(5)
***G*** = 1/2 (***F^T^. F*** − ***I***)(6)
***F*** = ***I*** + (∇***u***)**^T^**(7)
where *μ_s_* and *λ_s_* are Lamé constants, which are related to Young’s modulus and Poisson’s ratio of the material, respectively, **∑** is the second Piola–Kirchhoff stress tensor, ***G*** is the Green–Lagrangian strain tensor, and ***I*** is the second-order identity tensor.

### 2.3. Numerical Setup

Figure 2 shows a schematic of the 3D computational domain, with the origin located at the center of the leading edge. A fluid domain size of 3.5*Lx* × 3*Lx* × 3*Lx* is chosen along the *x*, *y*, and *z* in the thrust, spanwise, and vertical directions, respectively. These boundaries have been concluded to be large enough to have negligible effects on the fluid motion induced by the caudal fin. In Figure 2, the caudal fin of case 2 is illustrated as the solid domain; it has a chord length of *Lc* = 24 mm in the *x* direction and a thickness of 1 mm in the *z* direction. In the *y* direction, only half of the span (i.e., 20 mm for half of the caudal fin) is simulated using the symmetry plane boundary condition, which is a basic type of boundary in OpenFOAM for both fluid and solid domains. A structured mesh with the finest grids around the caudal fin’s surface is adopted to discretize the computational domain. The number of hexahedral cells in the mesh system is approximately 0.20–0.24 million, depending on the shape of the caudal fin.

Both the front surfaces of the fluid and solid domains are set to have *symmetryPlane* boundary conditions. For all patches on the side, top, and bottom surfaces, the *totalPressure* and *zeroGradient* boundary conditions are used for the pressure and velocity, respectively. At the interface coupled with the surface of the caudal fin, *movingWallVelocity* and *zeroGradient* boundary conditions are imposed for the velocity and pressure, respectively. For the solid domain, the vertical motion of the caudal fin is prescribed at the leading edge of the caudal fin, and the interface of the caudal fin surface is set to *tractionDisplacement* boundary condition for the displacement. 

Based on the experimental work of Baba et al. [27], the non-dimensional flapping frequency of each caudal fin is chosen to be around the natural frequency of each caudal fin, 0.6 < *f** < 1.5. Note that the natural frequency of each caudal fin varies with the fin shape, and the non-dimensional frequency *f** is the ratio of forcing frequency *f* to the corresponding natural frequency of each caudal fin *f_n_*. The density and kinematic viscosity of the fluid are *ρ* = 1.0 × 10^3^ kg/m^3^ and *υ_f_* = 1.0 × 10^−6^ m^2^/s, respectively, for the simulation of the caudal fin’s motion in stationary water. The flapping amplitude of the leading edge for all cases is *A_r_* = 3 mm. The caudal fin, which is made of silicone material, has a modulus of elasticity of 5.6 MPa and a density of 1280 kg/m^3^.

### 2.4. Validation and Grid Independence Study

To validate the current numerical method, simulations of a symmetric caudal fin with different trailing-edge shapes flapping at a wider range of non-dimensional frequencies, 0.6 < *f** < 1.5, in a stationary fluid were carried out. In the current study, the results of trailing edge amplitude and phase lag between the leading and trailing edge motion are compared with the experiments for three different caudal-fin shapes. The motion of the trailing edge is illustrated by its vertical displacement *z_t_* as;
*z_t_* = *A_t_* sin (2π*ϕ_t_*),      *ϕ_t_* = *ϕ* − *ϕ*_lag,t_,(8)
where *ϕ*_lag,t_ is a phase lag between the motion of the trailing edge and the leading edge of the caudal fin, and *A_t_* is the amplitude of the trailing edge. Note that the displacement *z_t_* is represented by the motion of the trailing edge at the tip of the side edge in each caudal fin (point *T* in Figure 1). Figure 3 shows the amplitude of the trailing edge *A_t_* and phase lag *ϕ*_lag,t_ of all cases for both the simulation and the experiment. It is clear that the present results are approximately the same as the experimental results.

In order to perform a grid independence study, calculations are carried out for three different resolutions of computational mesh in the fluid domain—coarse mesh, medium mesh, and fine mesh. The coarse, medium, and fine meshes have 110,809 cells, 221,611 cells, and 447,291 cells, respectively. The current test case is based on the flapping frequency *f** ≈ 1 for the deeply forked caudal fin of case 3. The details of the grid independence study and the validation of the numerical method have been presented in our previous work [34] for the square caudal fin of case 1. Figure 4 shows the effect of three different mesh resolutions on the displacement of the trailing edge. The results illustrate that the difference in the maximum value of trailing edge displacement is about 1.65% between the coarse and the medium mesh and 0.06% between the medium and the fine mesh. Therefore, the medium mesh is used for all simulations in the present study.

## 3. Results and Discussion

The motion of each caudal fin is characterized by the non-dimensional flapping frequency *f** = *f*/*f_n_*, where *f* is the forcing frequency, and *f_n_* is the natural frequency of the caudal fin. The natural frequency, *f_n_*, of each caudal fin is defined as the frequency at which the phase lag of the trailing edge relative to the leading edge of the caudal fin reaches π/2. The natural frequency of each caudal fin varies with the fin shape, and the values of *f_n_* in each caudal fin are *f_n_* = 1.5, 2.3, and 3.6 Hz for cases 1, 2, and 3, respectively. In order to clarify the relationship between the fin shape and the frequency dependence of its hydrodynamic performance, the simulations are carried out for a wider range of non-dimensional frequencies, 0.6 < *f** < 1.5, as in the experiments.

### 3.1. Hydrodynamic Performance

The thrust force, *F_T_*, in the *x*-direction is calculated using the following equation.
*F_T_* = 0.5 *C_T_*
*ρ*
*S*
*U*^2^(9)
where *C_T_* is the coefficient of thrust, *U* is the reference velocity, *ρ* is the fluid density, and *S* is the surface area of the caudal fin. The thrust force, *F_T_*, and the thrust coefficient, *C_T_*, can be calculated using the function tool of forces and force coefficients provided in OpenFOAM. Since there is no free stream velocity, the reference velocity is taken as the maximum translational velocity induced by the leading edge of the caudal fin, *U* = 2 π *f A*_r_. The reference velocity, *U*, in each case will vary with the forcing frequency, *f*, of each case. The surface area of each caudal fin is provided in Table 1. The variation of the mean thrust force with *f** for different fin shapes is shown in Figure 5. These results are obtained by averaging the instantaneous values of each thrust force over a heaving cycle.

Figure 5 illustrates that the thrust is increasing with *f** in a gradual manner in each case. Thrust becomes higher around the natural frequency and reaches a maximum at a certain value of *f** for case 1 and case 2. After reaching at a certain value of *f**, the mean thrusts of case 1 and case 2 decrease gradually. This value of *f** depends on the caudal-fin shape. The maximum thrust takes place at *f** = 1.2 and *f** = 1.3 for case 1 and case 2, respectively. For case 3, we find that thrust force increases monotonically as *f** increases in the frequency range used in this study. According to Figure 5, the trends of the variation in mean thrust force with *f** are observed to be consistent with those in Baba et al. [27] for all cases.

To compare the hydrodynamic performances in all cases, the results are presented in terms of the non-dimensional forms, in which the thrust coefficients and the hydrodynamic power input are defined as;
*C_T_* = 2*F_T_*/*ρ*
*S*
*U*^2^(10)
*C_P_* = 2*P*/*ρ*
*S*
*U*^3^(11)
where *P* is the instantaneous power input for initiating the motion of the caudal fin. The instantaneous power input, *P*, can be calculated from the product of the instantaneous forces, *F_Z_*, oscillating in the vertical direction and the flapping velocities of the leading edge of the caudal fin *z_r_*°.
*P*(*t*) = −*F_Z_*(*t*) *z_r_*° (*t*)(12)

The propulsive efficiency is calculated as follows:*η* = *C_T_*/*C_P_*(13)

The variations in the mean thrust coefficient and the mean power coefficient with *f** are shown in Figure 6 and Figure 7, respectively, for all cases. The thrust coefficient increases gradually up to a peak around the natural frequency and then gradually decreases with increasing *f**. The maximum value of *C_T_* in each caudal fin is found at the frequency ratio that provides the maximum amplitude of the trailing edge. Hence, the location of the peak in the thrust coefficient is well correlated with the location of maximum displacement of the trailing edge in each caudal fin. This suggests that the maximum thrust force is mostly achieved at the frequency of maximum amplitude in the fin’s motion. This is consistent with our previous finding [31] for a heaving rectangular plate, in which the maximum thrust force coefficient was related to the maximum non-dimensional amplitude of the trailing edge. At this frequency ratio, the mean thrust coefficient is about 1.033 in case 1, 0.899 in case 2, and 0.813 in case 3. Comparison among the three cases indicates that case 1 has the largest thrust coefficient in the frequency ranges below the natural frequency. Specifically, case 1 has largest value of *C_T_* when *f** < 1.2, while case 3 has a slightly larger value of *C_T_* when *f** > 1.2.

According to Figure 7, it is noticeable that the mean power coefficient in all cases decreases significantly as *f** decreases. As indicated in our previous study [31], the higher flapping frequency reduces the input power required to initiate the motion of the caudal fin. Comparing the three cases, significant differences are observed in the input power taken by the caudal fin. It seems that the caudal fin with a larger planform area requires more input power for initiating the motion of the caudal fin.

The propulsive efficiency’s correspondences to the thrust and power coefficients are shown in Figure 8 for the same range of *f**. The frequency characteristics of propulsive efficiency in case 1 and case 2 look similar. The efficiencies of these two cases increase gradually up to a peak of around 1.2 < *f** < 1.3 and then decreases gradually with increasing *f**. The maximum values of efficiency for case 1 and case 2 are 0.145 and 0.148, respectively. These two cases are more efficient than case 3 when *f** < 0.9, while case 3 provides greatest efficiency among the three cases when *f** > 0.9.

Interestingly, the trends of propulsive efficiency in all cases are similar to those of thrust force through the selected frequency range, shown in Figure 5. Compared to the square trailing edge of case 1, the efficiency of case 3 is reduced in the lower frequency range, while the efficiency of case 3 is greatly improved in the higher frequency range. At the lower frequency range of *f** ≈ 0.5, the propulsive efficiency of case 3 is 0.060, which is about 34% and 35% lower than those of case 1 and case 2, respectively. On the other hand, at the higher frequency range of *f** ≈1.5, the propulsive efficiency of case 3 is about 0.216, which is about 64% and 53% higher than those of case 1 and case 2, respectively. It is observed that the deeply forked caudal fin is the most efficient in the higher frequency range above the natural frequency. 

A recent experimental study by Van Buren et al. [26] reported that the thrust and efficiency of the pitching panels generally improved as the trailing-edge shape was changed from concave to square and convex shapes. Zhang et al. [29] also found that a square shape is favorable for enhancing performance in the region of small bending stiffness. Their results are consistent with the current findings, where the square caudal fin has greater thrust coefficients and higher efficiency in the lower frequency range below the natural frequency of each caudal fin. However, in the higher frequency range, the deeply forked caudal fin provides the greatest thrust and highest efficiency. Therefore, both shape and stiffness are important in evaluating the best propulsive performance, as suggested by Feilich et al. [30].

To examine the details of the thrust characteristics and flow fields for all cases, we first select the frequency ratio around the natural frequency of *f** ≈ 1, at which the maximum thrust coefficient is generated by each case over the non-dimensional frequency ranges. Then, we select the frequency ratio above the natural frequency of *f** ≈ 1.5, where the deeply forked caudal fin of case 3 provides the largest thrust and highest propulsive efficiency.

Temporal variations of the thrust coefficients for all cases over a flapping cycle at the frequency ratio of *f** ≈ 1 and *f** ≈ 1.5 are illustrated in Figure 9 and Figure 10, respectively. The results are computed from a heaving cycle that reaches a periodic steady state. The two maximum peaks of *C_T_* in a heaving cycle are well related with the symmetric heaving motion of each caudal fin. In Figure 9, the effect of fin shape on the generation of instantaneous thrust force is clearly observed, where case 1 provides the largest fluctuation of *C_T_* in a flapping cycle. Around the selected value of the non-dimensional flapping frequency *f** ≈ 1, the values of propulsive efficiency in all cases are close to each other. This indicates that the decreasing rate of mean thrust coefficients from case to case is similar to that of the power coefficients around *f** ≈ 1. 

When examining the details of the thrust characteristics at the selected value of non-dimensional flapping frequency *f** ≈ 1.5, it is observed that the maximum positive values of the thrust coefficients in all cases are quite similar, as seen in Figure 10. It would seem that the maximum negative value of the thrust coefficient is creating a decrease in the thrust coefficient in each case.

### 3.2. Flow Structures

Figure 11a and Figure 12 show the comparison of the pressure coefficient, *C_P_* = *p*/0.5 ρ*U*^2^, on both sides of each caudal fin surface at a phase instant when a maximum value of the instantaneous thrust coefficient appears in each case during the flapping cycle of *f** ≈ 1. The corresponding vortex structures around the caudal fin are illustrated in Figure 11b. The vortex structures are visualized using an iso-surface of the normalized Q-criterion (Q* = Q/*U*^2^) colored by the pressure coefficient *C_P_*.

By comparing the pressure coefficient among the three cases, it is evident that both the low- and high-pressure regions around the surface of case 1 are larger than those of the other two cases. Consequently, the pressure difference between the two sides of the caudal fin in case 1 is largest among the three cases, which causes the generation of the largest thrust coefficient at the corresponding phase instant, as seen in Figure 9. 

Figure 13a shows a comparison of the pressure coefficient on both sides of each caudal fin surface at a phase instant when the minimum value of the instantaneous thrust coefficient appears in each case during the flapping cycle of *f** ≈ 1.5. The corresponding vortex structures are illustrated in Figure 13b.

Due to the pressure difference around the leading-edge region of each caudal fin, the caudal fin cannot create forward motion, and a decrease in thrust coefficients occur in each case at these phase instants. In addition, it would seem that an extra decrease in the negative thrust coefficients arises from the low-pressure circular region created near the trailing-edge region. A significant low-pressure circular region is observed near the trailing edge of case 1, which seems to contribute an extra decrease in the thrust coefficients in case 1 when compared with the other two cases. 

## 4. Conclusions

In order to investigate the hydrodynamic performances of caudal-fin shapes, three-dimensional numerical simulation was conducted using open-source CFD software. Three types of caudal fin with different trailing-edge shapes were considered: square, forked, and deeply forked caudal fins. The leading edge of each caudal fin was forced to oscillate vertically in a water tank with zero free-stream condition, as in the experiments. Computations were conducted for a wide range of non-dimensional frequencies, both below and above the natural frequency of each fin type (0.6 < *f** < 1.5). A good quantitative agreement was observed between the computational and the experimental results in terms of the trailing edge amplitude and the phase lag of the trailing edge relative to the leading edge of each caudal fin.

The numerical results show that the amount of forking in the geometry of a caudal fin has significant effects on its hydrodynamic performance. A comparison of the thrust coefficients shows that the square caudal fin has a greater thrust coefficient in the non-dimensional frequency range of 0.6 < *f** < 1.2, while the deeply forked caudal fin generates higher thrust when 1.2 < *f** < 1.5. In term of propulsive efficiency, the square caudal fin is more efficient when 0.6 < *f** < 0.9, while the propulsive efficiency of a deeply forked caudal fin is significantly enhanced when 0.9 < *f** < 1.5. Based on our results, the deeply forked caudal fin has greater thrust coefficients and a higher propulsive efficiency in the frequency range higher than the natural frequency of each caudal fin. The performance results suggest that in the range of the non-dimensional frequency used in this study, changing the caudal-fin shape causes significant changes in the frequency characteristics of thrust coefficient and propulsive efficiency. Our results are helpful in understanding the hydrodynamic performance of various caudal-fin shapes and provide a suitable choice of caudal-fin shape for robotic underwater vehicles.

## Figures and Tables

**Figure 1 biomimetics-09-00445-f001:**
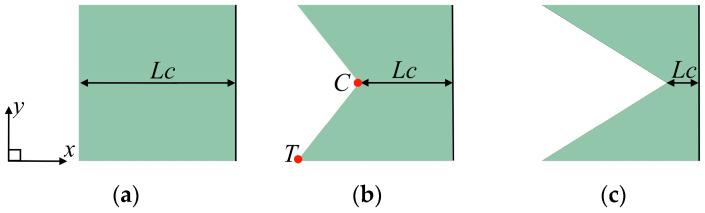
Three types of caudal fin with different trailing edges: (**a**) case1; (**b**) case 2; (**c**) case 3. The leading edge of caudal fin is indicated by the black line.

**Figure 2 biomimetics-09-00445-f002:**
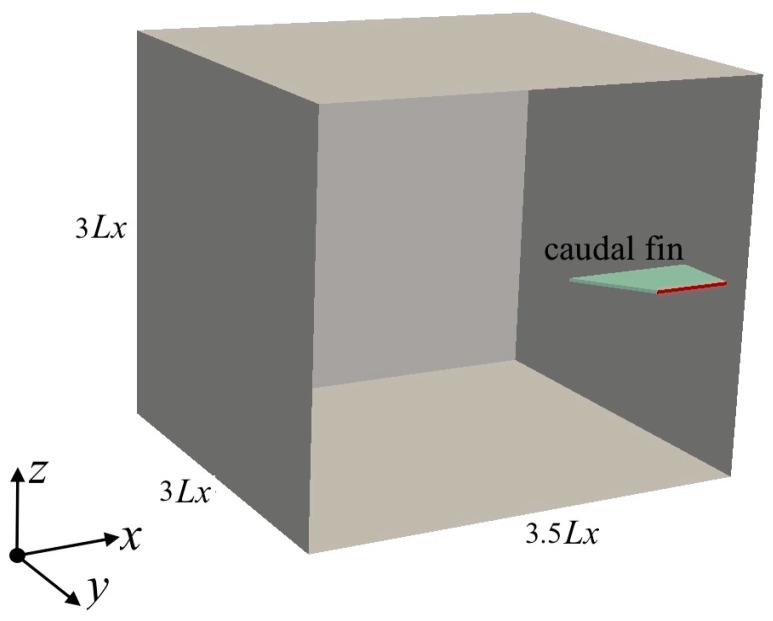
Computational domain. The symmetry plane of the caudal fin is indicated by the red line. The origin of the coordinate is at the center of the leading edge of the caudal fin.

**Figure 3 biomimetics-09-00445-f003:**
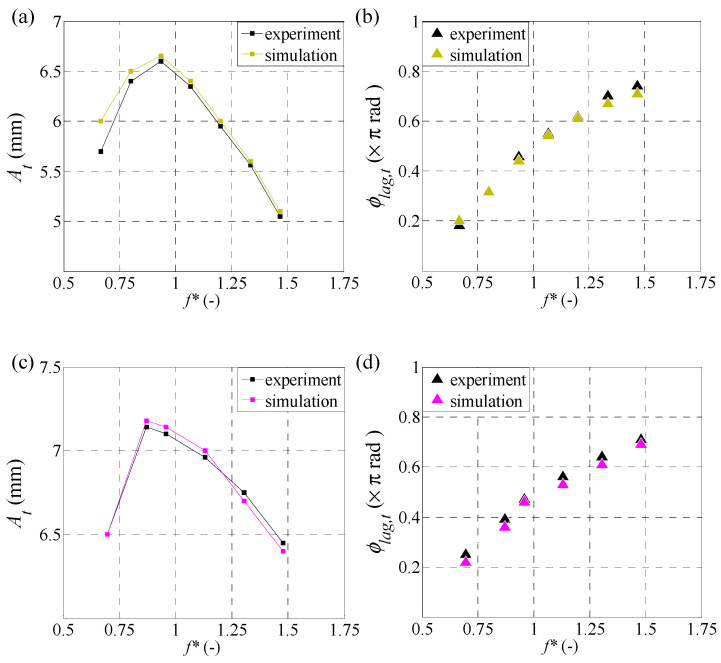
Comparison of amplitude *A_t_* and phase lag of trailing edge *ϕ*_lag,t_ between the simulation and experiment over a range of non-dimensional frequency *f** for different caudal-fin shapes: (**a**,**b**) case 1; (**c**,**d**) case 2; (**e**,**f**) case 3.

**Figure 4 biomimetics-09-00445-f004:**
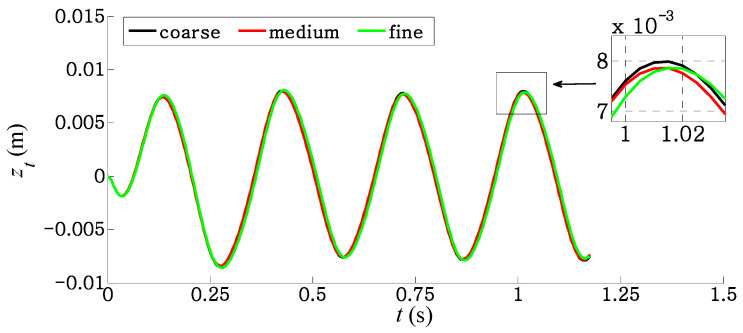
Displacement of trailing edge for three different mesh resolutions of case 3.

**Figure 5 biomimetics-09-00445-f005:**
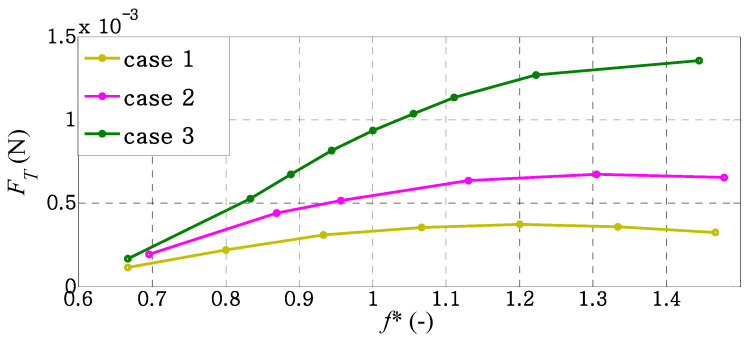
Mean thrust variation with *f** for different caudal-fin shapes.

**Figure 6 biomimetics-09-00445-f006:**
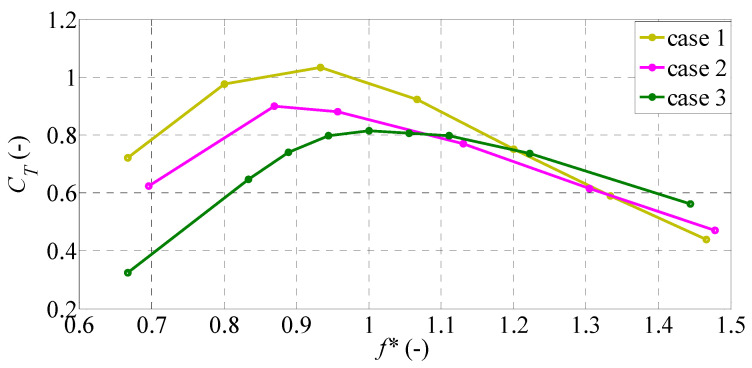
Mean thrust coefficient variation with *f** for different caudal-fin shapes.

**Figure 7 biomimetics-09-00445-f007:**
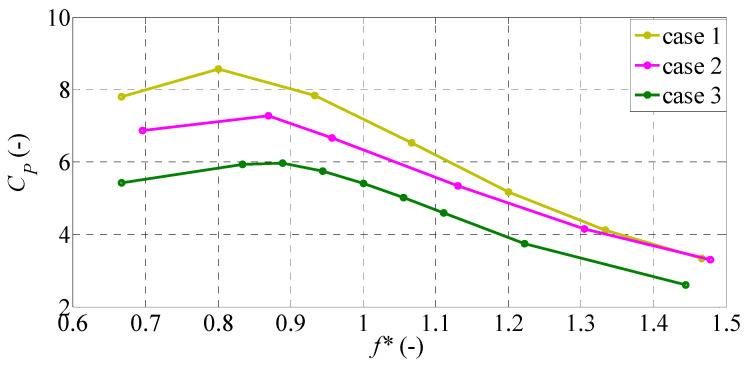
Mean power coefficient variation with *f** for different caudal-fin shapes.

**Figure 8 biomimetics-09-00445-f008:**
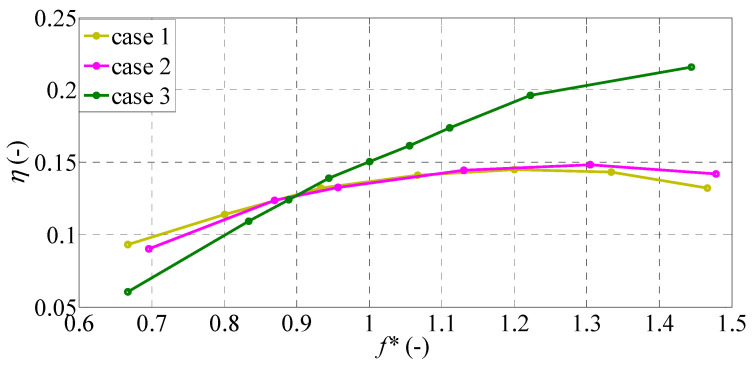
Propulsive efficiency variation with *f** for different caudal-fin shapes.

**Figure 9 biomimetics-09-00445-f009:**
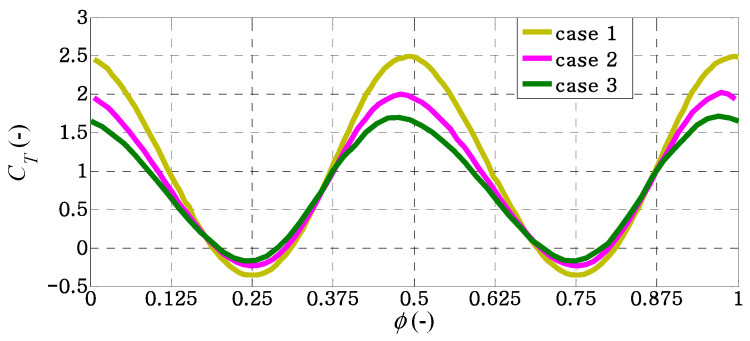
Instantaneous thrust coefficients for different caudal-fin shapes in a flapping cycle of *f** ≈ 1.

**Figure 10 biomimetics-09-00445-f010:**
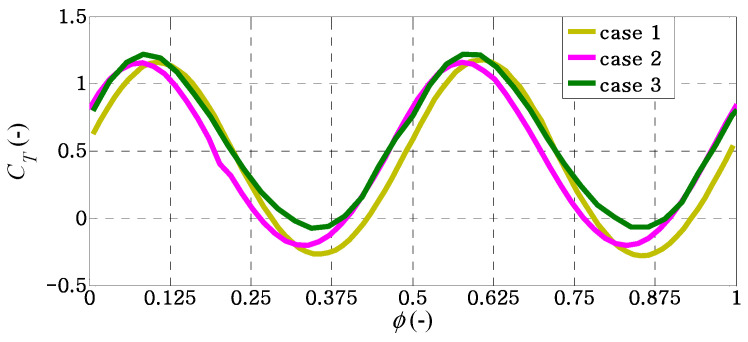
Instantaneous thrust coefficients for different caudal-fin shapes in a flapping cycle of *f** ≈ 1.5.

**Figure 11 biomimetics-09-00445-f011:**
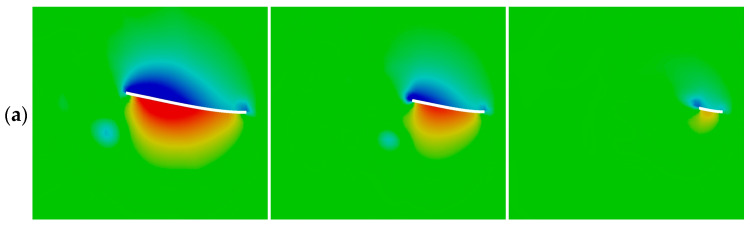
(**a**) Pressure coefficients on the mid-span section of the caudal fin and (**b**) the corresponding vortex structures at a phase instant when a peak value of instantaneous thrust coefficient appears in each case for *f** ≈ 1.

**Figure 12 biomimetics-09-00445-f012:**
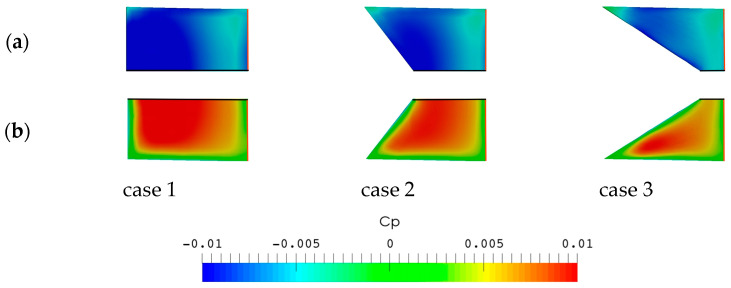
Pressure coefficients on the (**a**) top and (**b**) bottom surfaces of the caudal fin at a phase instant when a peak value of instantaneous thrust coefficient appears in each case for *f** ≈ 1 (the mid-span sections and leading edges are shown as black and red lines, respectively).

**Figure 13 biomimetics-09-00445-f013:**
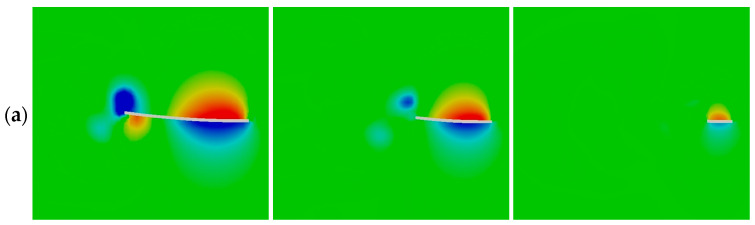
(**a**) Pressure coefficients on the mid-span section of the caudal fin and (**b**) the corresponding vortex structures at a phase instant when a minimum value of instantaneous thrust coefficient appears in each case for *f** ≈ 1.5.

**Table 1 biomimetics-09-00445-t001:** Geometric parameters.

Type	*Lx* [m]	*Lc* [m]	*S* [m^2^]	*AR* [-]
case 1	0.04	0.040	0.00160	1.00
case 2	0.04	0.024	0.00128	1.25
case 3	0.04	0.008	0.00096	1.70

## Data Availability

Please contact the corresponding author for the data presented in this paper.

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
