# Peer review of "Numerical Study on the Hydrodynamic Performance of a Flexible Caudal Fin with Different Trailing-Edge Shapes"

_biomimetics, 2024, doi:10.3390/biomimetics9070445_

Round 1

Reviewer 1 Report

Comments and Suggestions for Authors

This paper studies the influences of caudal fin shape on hydrodynamic performances. It’s found that the deeply forked caudal fin has greater thrust coefficients and higher propulsive efficiency in the frequency range higher than the natural frequency of each caudal fin. But the paper still lacks consideration in the following areas.

1.     In the first part of your paper, you introduce the current studies of the unsteady flow fields induced by the oscillation of caudal fin with five paragraphs, then you introduce the content of your own paper in the last paragraph. But the first five paragraphs are logically chaotic. You can start the narrative from three aspects: shape, kinematics and material flexibility, just like you have mentioned in your first paragraph.

2.     In the second part of your paper, I recommend you give the associated governing equations instead of just mentioning the work of other people.

3.     In the section 2.2 of your paper, you use half of the span to simulate in the y direction, but it’s uncertain whether this method is accurate and reliable. Especially, the grid size of your model is not very large (only 0.2 million). You should give more reasons and principles of this simplification and the boundary condition of the simplified cross section should be provided.

4.     The paper lacks grid independence verification, and the influence of grids on the three type fins needs to be considered.

5.     There are some errors in the title of Figure 3. The title only has (a)-(c), while the figure has (a)-(f). The image size of the three situations in Figure 9 (a) is inconsistent.

6.     In the third part of your paper, you give many results and conclusions. So I recommend you divide this part into several sub sections.

Comments on the Quality of English Language

Moderate editing of English language required

Reviewer 2 Report

Comments and Suggestions for Authors

Presented paper discussing hydrodynamic performance of the flexible caudal fin by using numerical study. Results of the paper gives good background for further study of such propulsion system for underwater robots. Paper contains several issues that should be solved before it can be published:

Introduction – please expand section with the focus on the previous numerical studies, published in the scientific literature;

Line 91 – Please remove Text “Simulations are conducted with open-source CFD software OpenFOAM”. This information already presented in the beginning of Methodology section;

Please add more information concerning mechanical properties of caudal fin that was chosen for simulations;

Please add the discussion section to discuss results especially in comparison with presented in the literature data to highlight novelty of your research.

In conclusion section please more highlight future application of received results.

Reviewer 3 Report

Comments and Suggestions for Authors

This manuscript presents numerical simulation results of fluid mechanics around a flexible caudal fin. The computed results are reliable as they agree with the experimental results in a reference paper. It is however, difficult to recommend this paper to proceed to further publication steps for the following reasons.

(1)   Although this manuscript conducted the simulations for the three different configurations of trailing edges, little new information is conveyed from the computational results. The fact that the different shapes of trailing edge produce different vibratory motions has been well known.

(2)   Why were the three types of trailing edge (Fig. 1) chosen? Are these three likely to exhibit better performance than others?

Comments on the Quality of English Language

The manuscript should be edited by natives.

Round 2

Reviewer 2 Report

Comments and Suggestions for Authors

Paper can be accepyed for publication in present form

Reviewer 3 Report

Comments and Suggestions for Authors

The revisions have sufficiently improved the manuscript to the acceptable state.

Comments on the Quality of English Language

The English writing maight be refined futhermore before going to the next step.